# Noninvasive liquid diet delivery of stable isotopes into mouse models for deep metabolic network tracing

Ramon C. Sun[1], Teresa W.-M. Fan [1,2], Pan Deng[1], Richard M. Higashi[1,2], Andrew N. Lane[1,2], Anh-Thu Le[3], Timothy L. Scott[1], Qiushi Sun [1], Marc O. Warmoes[1] & Ye Yang[1,2]

Delivering isotopic tracers for metabolic studies in rodents without overt stress is challenging. Current methods achieve low label enrichment in proteins and lipids. Here, we report non-invasive introduction of $^{13}C_6$-glucose via a stress-free, ad libitum liquid diet. Using NMR and ion chromatography-mass spectrometry, we quantify extensive $^{13}C$ enrichment in products of glycolysis, the Krebs cycle, the pentose phosphate pathway, nucleobases, UDP-sugars, glycogen, lipids, and proteins in mouse tissues during 12 to 48 h of $^{13}C_6$-glucose feeding. Applying this approach to patient-derived lung tumor xenografts (PDTX), we show that the liver supplies glucose-derived Gln via the blood to the PDTX to fuel Glu and glutathione synthesis while gluconeogenesis occurs in the PDTX. Comparison of PDTX with ex vivo tumor cultures and arsenic-transformed lung cells versus xenografts reveals differential glucose metabolism that could reflect distinct tumor microenvironment. We further found differences in glucose metabolism between the primary PDTX and distant lymph node metastases.

---

[1] Center for Environmental and Systems Biochemistry, University of Kentucky, 789S. Limestone St., Lexington, KY 40536, USA. [2] Department of Toxicology and Cancer Biology and Markey Cancer Center, University of Kentucky, 789S. Limestone St., Lexington, KY 40536, USA. [3] Department of Surgery, University of Kentucky, 800 Rose St, Lexington, KY 40536, USA. Correspondence and requests for materials should be addressed to T.W.-M.F. (email: twmfan@gmail.com)

Stable isotope tracer approaches, here termed Stable Isotope Resolved Metabolomics (SIRM) are becoming widely used for studying cancer metabolism[1, 2]. These approaches not only improved understanding of known metabolic phenotypes in cancer cells such as enhanced glycolysis (the Warburg effect[3]) and glutaminolysis[1, 4], but also revealed novel metabolic reprogramming crucial to tumor growth or survival to enable discovery of new therapeutic targets[2, 5, 6]. Most tracer studies have been performed in vitro using cell lines, which lack the tumor microenvironment and 3D architecture that may be crucial to recapitulate the transcriptional and metabolic programs in vivo. However, in vivo tracer studies remain challenging, due in part to the overt stress induced by animal handling with existing techniques, which also limits the duration of tracer delivery.

There are several established means of administering [13]C-glucose tracers in vivo, including bolus injections via the tail vein[7] and continuous infusion via cannulation[8, 9]. These methods may suffer from the well-documented effects on metabolism due to anesthesia[10, 11] and/or physical trauma[12, 13], thereby confounding the metabolic signatures of tumors. Although these invasive methods can achieve high [13]C-glucose levels in mouse plasma, tracer experiments are transient in nature (e.g., 30–150 min) to reduce stress responses induced by anesthesia or physical constraint. As such, [13]C labeling is often limited to metabolites of the faster turnover pathways such as glycolysis and the Krebs cycle[7, 8] but not the metabolome of more extended pathways such as de novo synthesis of lipids, proteins, and nucleotides. A better method of introducing stable isotope tracers is needed that avoids overt stress, while enabling long-term tracer administration for deeper pathway coverage in vivo.

Here, we report a non-invasive method of administering [13]C6-glucose to mouse models via liquid diet feeding to achieve deep metabolic network coverage. We have applied this method to non-small cell lung cancer patient-derived tumor xenografts and arsenite-transformed lung cell xenografts to assess the influence of the microenvironment on cancer cell metabolism.

## Results

**A liquid diet for in vivo [13]C enrichment of metabolites.** Here we describe a new method to introduce [13]C-glucose via an ad libitum liquid diet that is stress-free, highly reproducible, and achieves relatively high enrichment of complex carbohydrates, nucleotides, lipids and proteins in major organs of NOD/SCID/ Gamma (NSG) mice in <24 h. We designed a liquid diet formula in which glucose is the primary source of carbohydrate (Fig. 1a). Mice were habituated to an unlabeled glucose liquid diet for 2 days to achieve stable daily food intake (Fig. 1b). On day 3, the dietary carbohydrate was replaced with [13]C6-glucose. On the basis of the daily mouse-feeding pattern (Fig. 1c), we supplied the enriched liquid diet at 1600 hours, followed by necropsy at 1000 hours on the next day. Mouse organs were dissected, flash frozen in liquid nitrogen (LiqN2), and extracted for polar metabolites, lipids and proteins, then analyzed by NMR and mass spectrometry.

**Isotopic steady state is achieved in livers of [13]C-fed mice.** To investigate the approach to isotopic steady state in NSG mice fed with the [13]C6-glucose liquid diet, we performed a time course study by habituating 15 mice to an unlabeled glucose liquid diet for 48 h to achieve stable daily food intake, followed by fasting for 6 h then switching to the [13]C6-glucose diet (Fig. 1b). At 0, 12, 36, and 48 h after the switch, the mice were killed, followed by organ dissection and flash freezing of organs in LiqN2. Metabolites extracted from the liver of each mouse were analyzed by ion chromatography coupled to ultra high-resolution Fourier transform MS (IC-UHR-FTMS) to profile metabolites and their [13]C labeling patterns (Fig. 2).

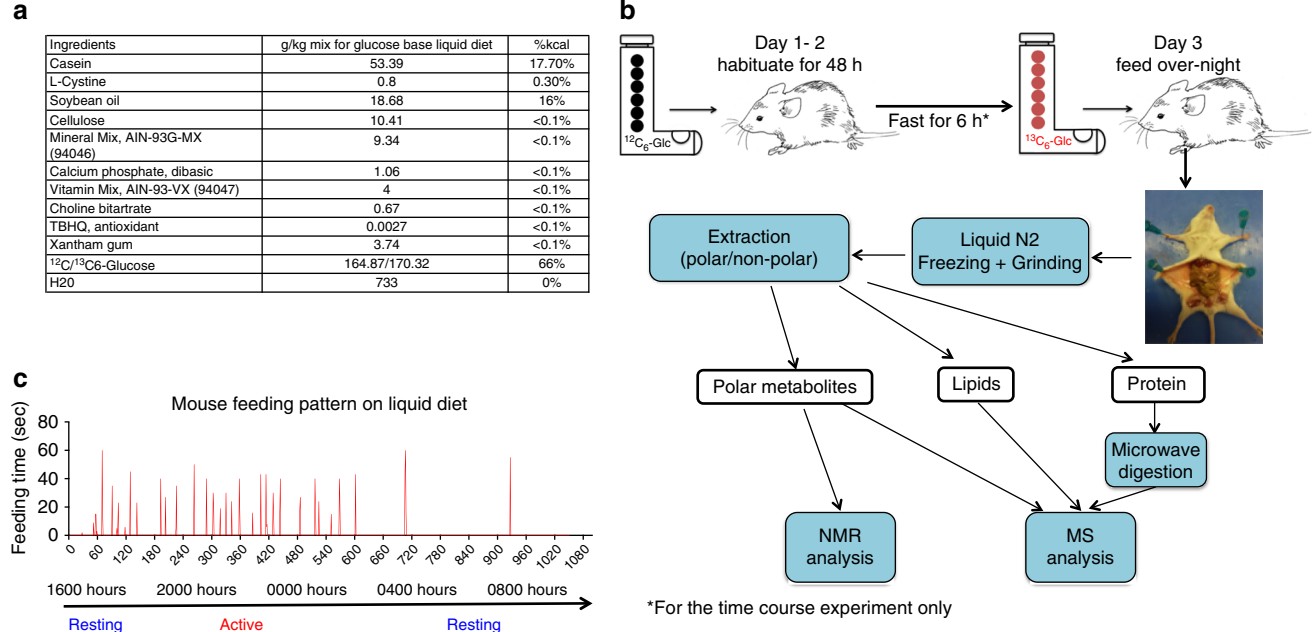

**Fig. 1** Procedure for tracing glucose metabolism in vivo. **a** Composition of liquid diet used for [13]C6-glucose tracer feeding (presented as g/kg and %kcal for each ingredient). The constituted liquid diet provides ca. 1000 kcal/l and a 20 g mouse is expected to consume 12.6 kcal a day. **b** Experimental workflow for tracking glucose metabolism in vivo. Mice were fed ad libitum [13]C6-glucose enriched diet for 18 h, after which blood was collected, and organs were dissected, snap frozen and pulverized in liquid nitrogen. Samples were then extracted into three fractions: polar, protein, and lipids. Each fraction was analyzed by NMR and/or Mass Spectrometry as described in the Methods. **c** Feeding pattern of an NSG mouse over 18 h

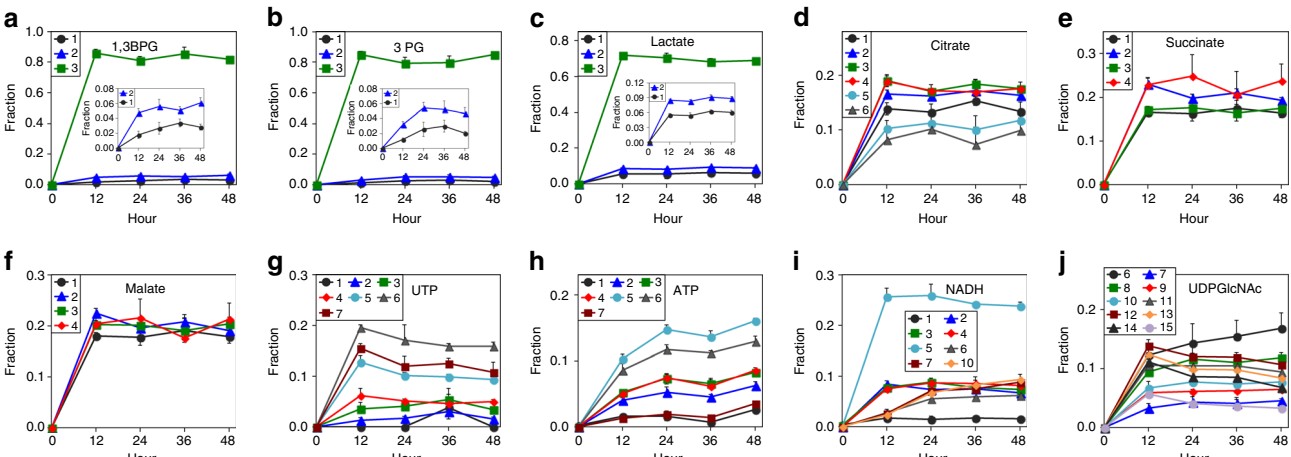

**Fig. 2** Time courses of $^{13}$C enrichment of central metabolites in mouse livers fed with $^{13}C_6$-glucose enriched liquid diet. WT NSG mice were necropsied at 0, 12, 36, and 48 h ($n = 3$ at each time point) after feeding with the $^{13}C_6$-glucose liquid diet. Time courses of the fractional enrichments of $^{13}$C isotopologues (denoted as **1-15**, where the number denotes the number of $^{13}$C atoms present) of key metabolites extracted from the liver were determined by IC-UHR-FTMS. Values represent mean ± SEM ($n = 3$). Fractional enrichment in the $^{13}C_1$ to $^{13}C_3$ isotopologues (**1-3**) of the glycolytic products 1,3-bisphosphoglycerate (1,3-BPG), 3-phosphoglycerate (3-PG) and lactate (**a-c**, main panels and insets with expanded Y-scale for the $^{13}C_1$ and $^{13}C_2$ isotopologues) reached steady state by 12 h. Fractional enrichment of the $^{13}$C isotopologues in the Krebs cycle products citrate **d**, succinate **e**, and malate **f**, as well as pyrimidine nucleotide UTP **g** also reached steady state by 12 h. However the fractional enrichment in some $^{13}$C isotopologues of ATP (**1-7** in **h**), NADH (**7,10** in **i**), and uridine diphosphate-N-acetylglucosamine (UDP-GlcNAc; **6, 8** in **j**) continued to rise during the 48 h period of tracer feeding

The fractional enrichment in the $^{13}C_1$ to $^{13}C_3$-isotopologues of glycolytic metabolites 1,3-bisphosphoglycerate (1,3-BPG, Fig. 2a), 3-phosphoglycerate (3PG, Fig. 2b) and lactate (Fig. 2c), was essentially constant from 12 to 48 h in the liver, indicating that isotopic steady-state was reached by 12 h for glycolysis. This was also the case for the Krebs cycle in the liver, as indicated by the constancy of the fractional enrichment in all of the $^{13}$C-isotopologues of the Krebs cycle metabolites citrate (Fig. 2d), succinate (Fig. 2e) and malate (Fig. 2f) after 12 h. To determine whether isotopic steady state was reached for metabolites further down the pathways, we measured the fractional $^{13}$C enrichment of nucleotides and nucleotide sugars. All of the significant $^{13}$C isotopologues of UTP ($^{13}C_{1-7}$, Fig. 2g) and uridine diphosphate N-acetylglucosamine (UDPGlcNAc, $^{13}C_7$, $^{13}C_{9-15}$-isotopologues, Fig. 2j) reached isotopic steady state by 12 h except for the $^{13}C_6$- and $^{13}C_8$-isotopologues of UDPGlcNAc. UDP-GlcNAc is a sugar nucleotide synthesized from glucose via glycolysis, the pentose phosphate pathway (PPP), and the hexosamine and pyrimidine pathways[14]. It is involved extensively in intercellular signaling and metabolic regulation (via O-linked glycosylation)[15, 16], as well as synthesis of the structural components of the cytoskeleton[17, 18]. Similarly, the extent of $^{13}$C incorporation into many of the NADH isotopologues reached steady state by 12 h except for the $^{13}C_7$- and $^{13}C_{10}$-isotopologues (Fig. 2i). In comparison, the fractional $^{13}$C enrichment of all significant ATP isotopologues (Fig. 2h) continued to rise during the 48 h period, but was close to isotopic steady state by 24 h. These data show that the liquid diet-enabled tracer introduction is an effective method for studying the complex dynamics of nucleotide and nucleotide sugar biosynthesis in vivo, including the differential temporal behavior of their $^{13}$C isotopologues.

**Glucose metabolism differs in major organs of NSG mice.** For subsequent mouse experiments, the same liquid diet procedure as described above was employed except that the fasting step was omitted (Fig. 1b). To demonstrate reproducibility, we analyzed the fractional $^{13}$C enrichment in the glycolysis and Krebs cycle metabolites of 10 mouse livers using IC-UHR-FTMS. The

coefficients of variation were 0.044 (citrate), 0.069 (malate), 0.076 (succinate) (Krebs cycle metabolites), and 0.033 (fructose-6-phosphate, F6P), 0.036 (glucose-6-phosphate, G6P) and 0.034 (phosphoenolpyruvate, PEP) (glycolysis metabolites) (Supplementary Fig. 1a), demonstrating high reproducibility of the liquid diet method.

To corroborate the reproducibility of data by IC-UHR FTMS, we used $^1$H{$^{13}$C}-HSQC NMR to determine the $^{13}$C abundances of various metabolites in polar extracts of 5 major organs, namely liver (Supplementary Fig. 1b) lung (Supplementary Fig. 1c), brain (Supplementary Fig. 1d), kidney (Supplementary Fig. 1e), and heart (Supplementary Fig. 1f) from three mice. Although there was some variation in the eating patterns of individual mice, the NMR spectra overlaid well for these organs, except for the glycogen resonances in the liver (Supplementary Fig. 1b). In addition, organ-specific $^{13}$C enrichment patterns of metabolites were observed in these labeled mice. For example, $^{13}$C-glycogen was observed in lung (Supplementary Fig. 1c) and liver (Supplementary Fig. 1b) while $^{13}$C-enriched Gln, γ aminobutyrate (GAB) and N-acetyl-aspartate (NAA) were high in brain (Supplementary Fig. 1d), which are consistent with known organ-specific metabolism[19, 20]. Moreover, $^{13}$C enrichment in the ribose subunits of free nucleotides (5.9–6.3 ppm) was observed in all organs, and was the highest in kidney (Supplementary Fig. 1e). De novo synthesis of the purine and pyrimidine rings were evident from the $^{13}$C signals in the 7.8–8.6 ppm of the liver, lung, and heart spectra (Supplementary Fig. 1b, c and f) respectively. Altogether, these enrichment patterns indicated active PPP and de novo nucleobase synthesis.

We then analyzed metabolites and their $^{13}$C isotopologues derived from multiple metabolic pathways using IC-UHR-FTMS. $^{13}$C-enriched metabolites from glycolysis (Supplementary Fig. 2a–f), the Krebs cycle (Supplementary Fig. 2g–k), PPP (Supplementary Fig. 3), and sugar nucleotide pathways (Supplementary Fig. 4) were identified and their fractional enrichments were determined. Considerable mixing of $^{13}$C labels (especially the $^{13}C_3$ isotopologues) in glucose-6-phosphate (G6P) and fructose-6-phosphate (F6P) (Supplementary Figs. 2a and 3b) was evident in all five organs. As both liver and kidney are known

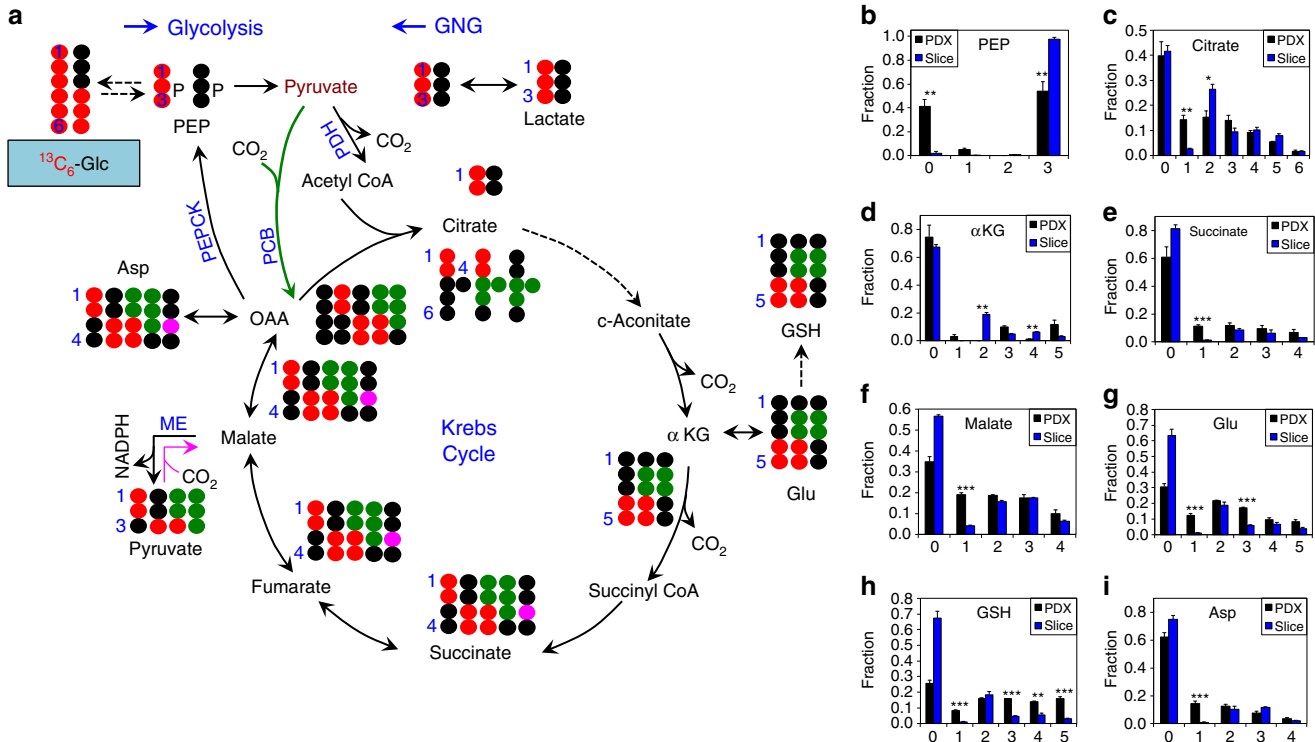

**Fig. 3** A PDTX model of NSCLC shows attenuated glycolysis and enhanced malic enzyme activity and use of non-Krebs cycle-derived αKG for Glu/GSH synthesis relative to the corresponding ex vivo tissue culture model. Mice bearing F0 PDTX ($n = 3$) from patient UK025 were fed the $^{13}C_6$-Glc enriched liquid diet for 18 h (cf. Fig. 1) before necropsy. Tumor tissues were dissected, weighed, and flash-frozen in liquid $N_2$. The freshly thin-sliced UK025 tumor tissues were cultured ex vivo in $^{13}C_6$-glucose for 24 h before flash-frozen in liquid $N_2$. Both sets of tissues were processed and analyzed by IC-UHR-FTMS as described in Methods. **a** Tracing of glucose carbon through glycolysis, gluconeogenesis (**GNG**), the Krebs cycle, malic enzyme (ME) reactions, and synthesis of Glu/GSH. Not all possible labeled metabolites are shown. Black circle: $^{12}$C; red circle, green circle: $^{13}$C from pyruvate dehydrogenase (PDH) and pyruvate carboxylase (PCB)-initiated Krebs cycle reactions, respectively; pink circle: $^{13}$C derived from ME reaction; PEPCK: PEP carboxykinase. **b-i** fractional enrichments in the isotopologues of phosphoenolpyruvate (PEP), citrate, α-ketoglutarate (αKG), succinate, malate, Glu, reduced glutathione (GSH), and Asp, respectively. The x-axis denotes the number of $^{13}$C atoms present in each compound. Values shown are mean ± SEM ($n = 3$). $^*$ $0.01 < P < 0.05$; $^{**}$ $0.001 < P < 0.01$; $^{***}$ $P < 0.001$, two-tailed $t$-test (see Methods)

to be active in gluconeogenesis (cf. Fig. 3a) and the PPP (Supplementary Fig. 3a), the mixed $^{13}$C labeling patterns observed for G6P and F6P were expected in these organs. The $^{13}$C mixing of G6P or F6P in other organs could arise from gluconeogenesis, the non-oxidative branch of PPP (cf. Supplementary Fig. 3a), and/or uptake of scrambled $^{13}$C-labeled glucose released by the liver or kidney into the circulating blood. If liver were the sole source of the scrambled isotopic species, then we would expect the fractional $^{13}$C enrichment of G6P and F6P to be lower in these organs than in liver, which was not the case (Supplementary Figs. 2a, 3b). We observed PPP activity in all 5 organs, with lung and liver being most active, as evidenced by extensive $^{13}$C enrichment (60-95%) in the intermediates ribose-5-phosphate (R5P, Supplementary Fig. 3c), erythrose-4-phosphate (E4P, Supplementary Fig. 3d) and S7P (Supplementary Fig. 3e). Thus, the PPP could variably contribute to the $^{13}$C scrambling in G6P and F6P. Moreover, fully $^{13}$C-labeled Krebs cycle metabolites (e.g., $^{13}C_6$-citrate, $^{13}C_5$-αketoglutarate (αKG), $^{13}C_4$-succinate, $^{13}C_4$-fumarate, and $^{13}C_4$-malate) dominated in the brain (Supplementary Fig. 2g–k), which is presumably produced from $^{13}C_6$-glucose via at least 4 turns of the canonical Krebs cycle activity and/or 2 turns of the cycle activity with anaplerotic pyruvate carboxylation. The production of these fully labeled species is consistent with the high Krebs cycle and anaplerotic pyruvate carboxylase (PCB) activity in the brain[21] and with the high demand for glucose as the major carbon source in the brain[22]. We further found extensive but variable $^{13}$C labeling in

UDP-GlcNAc (the 17-carbon sugar nucleotide synthesized via glycolysis, PPP, hexosamine, the Krebs cycle, and pyrimidine pathways, Fig. 4) including the fully labeled $^{13}C_{17}$ species, particularly in the brain (Supplementary Fig. 4d).

As citrate is the precursor of cytoplasmic acetyl-CoA used for fatty acid synthesis, the high degree of labeling in citrate/isocitrate (Supplementary Fig. 2g) suggests that the fatty acids and lipids may also be $^{13}$C-enriched. We therefore analyzed the lipid extracts by direct infusion nanoelectrospray (nESI) UHR-FTMS. Liver had the highest fractional enrichment of $^{13}$C labeled glycerolipids (lowest $^{13}C_0$ or **0** isotopologue, Supplementary Fig. 5), mainly in the fatty acyl chains ($^{13}C_{even}$, $^{13}C_{odd} > 3$) (Supplementary Fig. 5), consistent with liver's central role in lipid metabolism[23].

The high fractional $^{13}$C enrichment in Glu also raised the question of whether $^{13}$C-glucose-derived non-essential amino acids support de novo protein synthesis. We tracked this process by microwave-assisted acid hydrolysis of total protein, followed by derivatization of released amino acids with ethyl chloroformate and analysis using nESI-UHR-FTMS[24] (Supplementary Fig. 6a). We found the highest $^{13}$C enrichment (up to 10% in the liver) in proteinaceous amino acids to be $^{13}C_3$-Ala (**3**, Supplementary Fig. 6b), which is presumably derived from glycolysis. We also found that the fractional enrichment of the proteinaceous $^{13}C_3$-Asp (**3**, Supplementary Fig. 6c) was generally higher than that of $^{13}C_2$-Asp in the kidney, brain, and heart. The former isotopologue is likely to be derived from the pyruvate

carboxylation pathway (cf. Fig. 3 and ref. [25]). Furthermore, we noted that $^{13}C_3$-Ser (**3**, Supplementary Fig. 6e) incorporation into proteins was highest in the brain. These data indicate substantial de novo protein synthesis in all five organs during the 18 h of tracer feeding.

**PDX and ex vivo tissue cultures differ in glucose metabolism.** We then applied the method to track tumor-specific metabolic networks in NSG mice bearing the first generation (F0) patient-derived NSCLC tumor xenograft (PDTX) and compared them to those of the ex vivo tumor tissue slices procured from the same patient[26, 27]. Semi-quantitative $^1H\{^{13}C\}$-HSQC NMR analysis showed $^{13}C$ labeling in numerous metabolites both in the PDX tumor and the tumor tissue slices (Supplementary Fig. 7a), including nucleotide ribose, glycogen, glutathiones (GSH + GSSG), Gln, succinate, Glu, acetate and alanine. However, $^{13}C$-labeled lactate, Ala, glutathiones, Glu, Gln, succinate, and the ribose residue of UXP were more abundant in the PDTX than in the matched ex vivo tumor slices. These increases in $^{13}C$ abundance of metabolites in PDTX are in contrast to the reduced fractional enrichment in the $^{13}C_6$-glucose input from the host plasma (cf. **6** or $^{13}C_6$ isotopologue, Supplementary Fig. 9a). They could reflect the influence of the tumor microenvironment (TME), such as differences in glucose uptake and/or human versus mouse stroma.

PDTX and ex vivo tumor extracts were also analyzed by IC-UHR-FTMS to confirm and complement the NMR analysis. We observed high $^{13}C$ enrichment in metabolites of glycolysis (Supplementary Fig. 8), the Krebs cycle (Fig. 3), PPP (Supplementary Fig. 8g, h), and nucleotides/sugar nucleotides (Supplementary Fig. 8i–k) in both systems but their $^{13}C$ labeling patterns differed. Notably, lower fractional enrichment of $^{13}C_3$-PEP (Fig. 3b) and $^{13}C_3$-lactate (Supplementary Fig. 8f) was evident in the PDTX versus the ex vivo tumor tissues, which could be related to the dilution of the $^{13}C_6$-glucose input by other unlabeled glucose source(s) in the host plasma (Supplementary Fig. 9a) and/or attenuated glycolysis in the PDTX tissues. In addition, the total fractional enrichment in the $^{13}C$ scrambled isotopologues of G6P ($^{13}C_1$ to $^{13}C_5$; **1-5**, Supplementary Fig. 8a), F6P ($^{13}C_1$ to $^{13}C_5$; **1-5**, Supplementary Fig. 8b), lactate ($^{13}C_1$ to $^{13}C_2$; **1-2**, Supplementary Fig. 8f), and S7P ($^{13}C_1$ to $^{13}C_6$; **1-6**, Supplementary Fig. 8h) were much higher in the PDTX than in the ex vivo tumor slices, which could be related to the significant $^{13}C$ scrambling of $^{13}C_6$-glucose in the host plasma (Supplementary Fig. 9a). However, the $^{13}C$ scrambling patterns of plasma glucose cannot fully account for those of the products G6P and F6P in PDTX tissues since the extent of $^{13}C$ scrambling was higher in the tissue products than the plasma input. In addition, the fractional enrichment of the $^{13}C_3$-isotopologues (a marker of gluconeogenesis) of G6P and F6P in PDTX tissues was higher than that of the input glucose in the plasma (Supplementary Fig. 9a). Moreover, the $^{13}C$ scrambling patterns of G6P and F6P deviated from those of the PPP precursor S7P (Supplementary Fig. 8h). Altogether, these results pointed to the contribution of in situ gluconeogenesis in addition to the non-oxidative branch activity of PPP and hepatic gluconeogenesis to the $^{13}C$ enrichment patterns of G6P and F6P in PDTX tissues, as reasoned above. The much lower extent of $^{13}C$ scrambling in G6P, F6P, or S7P in ex vivo tumor slices suggested much reduced activity of all three pathways, which may again reflect a TME influence.

Tracking $^{13}C_6$-glucose metabolism via the Krebs cycle (Fig. 3a), we observed substantial fractional $^{13}C$ enrichment in the $^{13}C_2$- and $^{13}C_3$-isotopologues of citrate, malate, fumarate, and Asp of both PDTX and ex vivo human tumor tissues (Fig. 3), which is consistent, respectively, with the heightened activity of PDH and

PCB-initiated Krebs cycle (Fig. 3a) in human NSCLC tumors observed in-patient[2]. We also noted much higher extent of enrichment in $^{13}C_1$-malate, -fumarate, -Asp, and -succinate in PDTX versus ex vivo tumor tissues. These isotopologues could be derived from the malic enzyme (ME) reaction in reverse (denoted by pink circle in Fig. 3a). We further noted lower extent and different patterns of $^{13}C$ enrichment in αKG (Fig. 3d) versus its expected downstream products Glu (Fig. 3g) and GSH (Fig. 3h) in PDTXs. This result suggested the presence of additional $^{13}C$ labeled αKG to fuel Glu/GSH synthesis in PDTXs. We found comparable level and similar patterns of $^{13}C$ enrichment in plasma Gln of the host mice (Supplementary Fig. 9b), which was de novo synthesized from $^{13}C_6$-glucose, presumably in the liver. Thus, Glu and GSH synthesis in PDTXs could be largely fueled by Gln in the blood rather than from αKG synthesized in situ.

**Nucleotide and GSH synthesis increases in lymph node metastases.** During necropsy of the F2 generation of the above PDTX model, we observed development of metastatic lesions in the brachial lymph nodes, which had similar gross tumor morphology (Supplementary Fig. 10a, b) and histopathology (Supplementary Fig. 10c, d) as the matched primary PDTX. These lymph node lesions stained strongly for human nucleolar antigens (Supplementary Fig. 10d), which indicated their cells to be of human lineage. This provided an excellent opportunity to profile differences in metabolic activities between primary and metastatic lesions originating from the same human tumor using the $^{13}C$ liquid diet approach. After 18 h of feeding with $^{13}C_6$-glucose enriched liquid diet, the primary and metastatic lesions were processed and analyzed by IC-UHR-FTMS, as described in the Methods. This analysis provided the $^{13}C$ labeling patterns for relevant metabolites, both in terms of the pool size (Fig. 4) and fractional enrichment (Supplementary Fig. 11) of their $^{13}C$ isotopologues, to assess the capacity for glycolysis, PPP, the Krebs cycle, GSH metabolism, and nucleotide biosynthesis. As Fig. 4 and Supplementary Fig. 11 shows, the $^{13}C$ labeling patterns were similar between the primary tumor and lymph node lesions for metabolites of glycolysis (cf. $^{13}C_3$-lactate/pyruvate, Fig. 4b, l and Supplementary Fig. 11b, l), gluconeogenesis (cf. $^{13}C_3$-G6P, Fig. 4a and Supplementary Fig. 11a), and the Krebs cycle including the anaplerotic PCB (cf. $^{13}C_3$-citrate/Asp, Fig. 4c, g and Supplementary Fig. 11c, g), ME (cf. $^{13}C_1$-malate, Fig. 4f and Supplementary Fig. 11f) and canonical PDH-initiated reactions ($^{13}C_2$-citrate/Asp, Fig. 4c, g and Supplementary Fig. 11c, g). These data pointed to a similar capacity of these pathways in the two tumor types.

Interestingly, the lymph node lesions showed consistently higher levels of $^{13}C$ incorporation from $^{13}C_6$-glucose into the PPP (Fig. 4i), GSH (Fig. 4h), and nucleotide products (Fig. 4j, k) than the primary tumor while the $^{13}C$ fractional enrichment for these pathway products was comparable between the two tumor types (Supplementary Fig. S1i, h, j, k). These data suggest that the changes in capacity for these pathways in lymph node lesions was comparable when fueled by glucose or non-glucose source(s) since the pool sizes for the unlabeled ($^{13}C_0$) and $^{13}C$ labeled isotopologues of these pathway products were similarly elevated leading to little changes in their fractional enrichment. Thus, metastatic lesions appeared to have altered capacity for anti-oxidation and nucleotide biosynthesis relative to primary lung tumors, presumably to meet their survival and growth demands. These changes could also reflect the influence of tumor size, stage, and/or microenvironment on tumor metabolism.

**BAsT cell metabolism is sensitive to the TME.** To further probe if and how cancer cell metabolism is sensitive to the TME, we

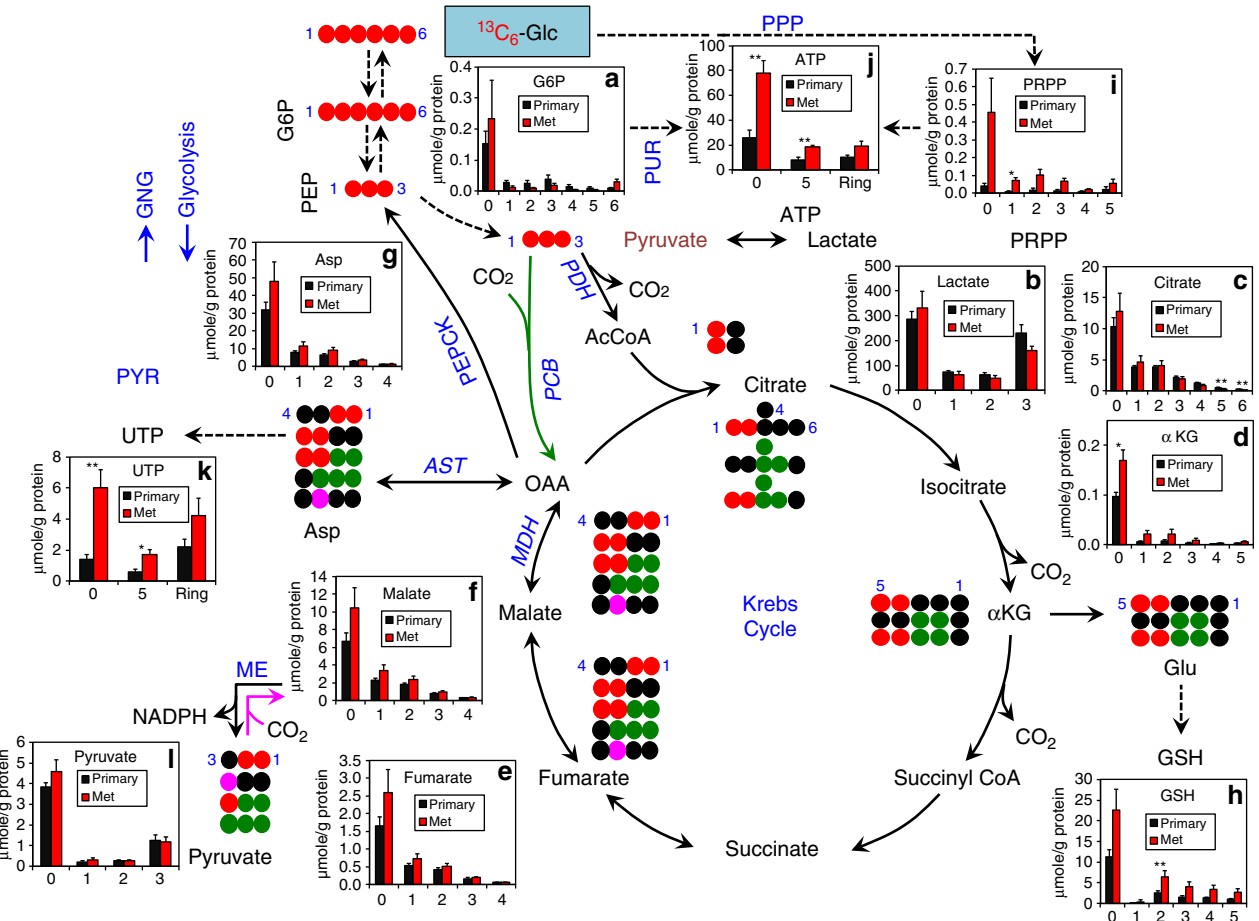

**Fig. 4** Lymph node metastases show enhanced PPP and biosynthesis of glutathione and nucleotides relative to the matched primary PDTX in the same mouse. Mice bearing F2 primary PDTX and metastatic lymph node lesions (Met) (from patient UK025) were fed the $^{13}C_6$-Glc enriched liquid diet for 18 h (cf. Fig. 1) before necropsy. Tissues were processed and analyzed by IC-UHR-FTMS, as described in Fig. 3. Glucose-$^{13}$C is traced through glycolysis, gluconeogenesis (**GNG**), PPP, the Krebs cycle, ME reactions, and synthetic pathways of GSH, purine nucleotides (**PUR**), and pyrimidine nucleotides (**PYR**). Not all possible labeled metabolites were shown. **a–l** μmole/g protein content of isotopologues of glucose-6-phosphate (G6P), lactate, citrate, αKG, fumarate, malate, Asp, GSH, phosphoribosyl pyrophosphate (PRPP), ATP, UTP, and pyruvate, respectively in primary tumor versus lymph node metastases (cf. Supplementary Fig. 11 for the fractional distribution of these isotopologues). The x-axis represents the number of $^{13}$C atoms present in each compound; for ATP and UTP, "ring" is the summed value for the $^{13}C_6$ to $^{13}C_9$ isotopologues. All other abbreviations and symbols are as in Fig. 3. Values shown are mean ± SEM (n = 5). * $0.01 < P < 0.05$; ** $0.001 < P < 0.01$; *** $P < 0.001$, two-tailed t-test (see Methods)

obtained arsenite-transformed human bronchial alveolar BEAS-2B (BAsT) cells (gift of Dr Y. Fondufe-Mittendorf, University of Kentucky) and xenografted these cells into NSG mice to compare glucose metabolism of the same cells in vitro versus in vivo. After 4 months of tumor growth, 3 mice were administered the $^{13}C_6$-glucose-based liquid diet for 18 h, followed by resection of tumor tissues, extraction for polar metabolites, and analysis by NMR and IC-UHR-FTMS. Parallel $^{13}C_6$-glucose-based SIRM experiments were performed on BAsT cells in vitro.

Supplementary Fig. 7b and 5 compare the central metabolic activity of BAsT cells as tumor xenograft versus in vitro monolayer cultures. We found multiple distinct features of the BAsT tumor xenografts relative to cells in vitro, which were consistent with: (1) more active glycogen synthesis (higher abundance of $^{13}$C-glycogen, (Supplementary Fig. 7b); (2) uptake of plasma glucose derived from hepatic gluconeogenesis (scrambled $^{13}$C patterns of fructose-1,6-bisphosphate (F1,6BP, **a**, pyruvate **b**, and lactate **c**, Fig. 5); (3) enhanced PCB-initiated Krebs cycle activity (elevated fractional enrichment in the $^{13}C_3$- and $^{13}C_5$-isotopologues of citrate **d** as well as $^{13}C_3$- and $^{13}C_4$-isotopologues of Asp **k**, Malate **h**, fumarate **g** and succinate **f**, Fig. 5) similar to the case of human NSCLC tumors in vivo;[2, 28]

(4) elevated ME activity in the reverse direction (higher fractional enrichment of $^{13}C_1$-isotopologues of malate, fumarate, succinate, and Asp); (5) utilization of non-tumor derived $^{13}$C-αKG (e.g., from hepatic $^{13}$C-Gln) for Glu synthesis (total $^{13}$C fractional enrichment of Glu **i** was higher than that of its precursor αKG (Fig. 5e); (6) more favorable non-oxidative PPP ($^{13}$C scrambling in the PPP intermediate S7P (Fig. 5l) was much more extensive). Except for (3), the rest of the metabolic features were akin to those of the PDTXs when compared with the ex vivo tissue counterparts (cf. Fig. 3, Supplementary Fig. 7a).

Moreover, as for the PDTX versus tissue slice case described above, the uptake of $^{13}$C scrambled glucose from the plasma (Supplementary Fig. 9c) could contribute to and complicate the interpretation of the $^{13}$C labeling patterns of tumor xenograft metabolites, particularly in terms of gluconeogenesis. However the $^{13}$C labeling patterns of plasma glucose (e.g., dilution of $^{13}C_6$- by unlabeled glucose, Supplementary Fig. 9c) could not account for the elevated $^{13}$C abundance of glycolytic (Lac/Ala) and Krebs cycle metabolites (Glu/succinate/Asp) (Supplementary Fig. 7b) and enhanced fractional enrichment of $^{13}$C-labeled Krebs cycle metabolites. Neither could they explain the distinct scrambled $^{13}$C labeling patterns of F1,6BP ($^{13}C_3$-isotopologue in particular,

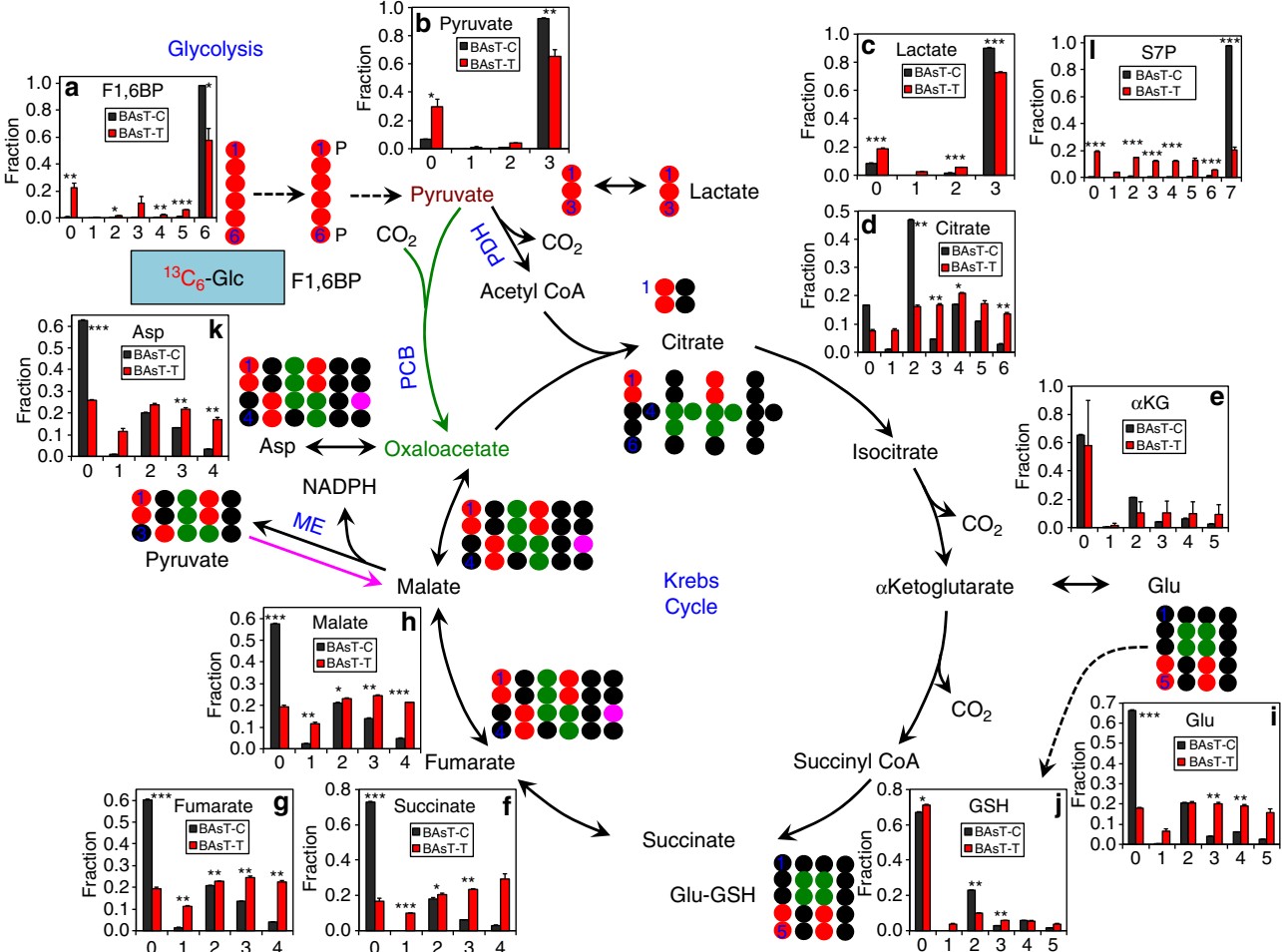

**Fig. 5** Central metabolic activity of arsenite-transformed BAsT cells is sensitive to the tumor microenvironment. Human bronchial alveolar BEAS2B cells were transformed by exposure to 1 µM $Na_2AsO_3$ for 18 weeks before subcutaneous xenograft of the transformed cells (BAsT cells) into NSG mice. After 4 months, mice were fed the $^{13}C_6$-Glc enriched liquid diet for 18 h (cf. Fig. 1) before necropsy. Tumor tissues were processed and analyzed by IC-UHR-FTMS as described in Fig. 3. A $^{13}C_6$-Glc tracer experiment was also performed for the BAsT cells as in vitro culture for 24 h in DMEM before extraction for polar metabolites, followed by IC-UHR-FTMS analysis (n = 3). Shown in **a–c**, **d–k**, and **l** are the fractional enrichments in the isotopologues of metabolites derived from glycolysis, the Krebs cycle, and the PPP, respectively. F1,6BP: fructose-1,6-bisphosphate; S7P: sedoheptulose-7-phosphate; BAsT-C (black square): in vitro cell cultures; BAsT-T (red square): tumor xenografts; all other abbreviations are as in Fig. 3. The x-axis represents the number of $^{13}C$ atoms present in each compound. Values shown were mean ± SEM (n = 3). $^*$ $0.01 < P < 0.05$; $^{**}$ $0.001 < P < 0.01$; $^{***}$ $P < 0.001$, two-tailed t-test (see Methods)

Fig. 5a) relative to those of the precursor S7P (Fig. 5l) or plasma glucose (Supplementary Fig. 9c) in tumor xenografts. The latter implicated gluconeogenic activity. Thus, our data suggested that TME in vivo could promote anaplerotic pyruvate carboxylation, non-oxidative branch of PPP, malic enzyme activity in the reverse or anaplerotic direction, and gluconeogenesis.

## Discussion

We have established a liquid diet that enabled non-invasive delivery of $^{13}C$ labeled glucose for in vivo metabolic studies. The demonstrated advantages of this approach include: (1) simple delivery of tracers without anesthesia or physical restraint; (2) high reproducibility with coefficient of variance ranging from 0.033 to 0.076 for the Krebs cycle and glycolytic metabolites, which will be beneficial for in vivo studies with small sample sizes such as early generations of PDTX; and (3) much deeper coverage of metabolic networks such that the turnover of lipids, nucleotides, and protein are observable, which has been beyond the reach of short-term, invasive methods of tracer administration. Altogether, these advantages represent a major step forward for

in vivo metabolic studies of animal models, particularly for examining the metabolic interactions of hosts and tumor xenografts. However, the liquid diet method may not be compatible with very short-term tracer studies due to possible variations in the feeding behavior. The specific method of choice for tracer administration will depend on the question being posed.

Our findings in two types of tumor xenografts are consistent with the metabolic scheme outlined in Supplementary Fig. 12, where the liver supplies glucose-derived Gln via the blood to the xenograft to fuel Glu or GSH synthesis and gluconeogenesis occurs not only in the liver but also in the xenograft. It is also notable that glycolysis was attenuated, glycogen synthesis was far more active, and the ME reaction was more operative in the anaplerotic direction in the tumor xenograft than in the corresponding monolayer cell culture. The differences in cancer cell metabolism observed for the in vivo versus ex vivo or in vivo versus in vitro models may reflect differences in multiple microenvironmental factors such as nutrient supply (e.g., high glucose and glutamine ex vivo and in vitro) and exchanges (via other organs in vivo versus culture media ex vivo and in vitro), growth factors (mouse serum versus FBS), tumor-associated

stroma (in vivo mouse versus ex vivo human). However, the PDX model also displayed many metabolic similarities to the corresponding ex vivo tumor tissue models; the latter metabolism in turn recapitulates many aspects of the in vivo tumor metabolism in human patients[2]. Moreover, the metabolic adaptations of human tumor cells in the mouse stroma could have important implications in tumor cells' ability to survive glucose-deficiency in vivo. For example, decreased glycolysis conserves glucose consumption, while enhanced glycogen synthesis and accumulation can serve as a glucose reserve. Increased glycogen stores have been shown to protect cancer cells from bioenergetic stress[29] and inhibition of glycogen degradation sensitized cancer cells to glucose starvation[30] or restricted their proliferation[31]. Gluconeogenesis can also supply glucose to support critical metabolic processes. Although it is not immediately apparent as to the benefit of enhanced ME activity, we surmise that this activity could facilitate the utilization of non-glucose carbon sources such as Gln to sustain the Krebs cycle activity under glucose deficiency[32]. Future $^{13}$C-Gln-based tracer studies in vivo would help verify this hypothesis.

In conclusion, stable isotope tracers-based mapping of metabolic networks is a powerful approach to studying cancer metabolism in vivo in tumor xenografts, particularly for probing the influence of mouse TME on metabolic reprogramming in human cancer cells. However, current in vivo labeling techniques face important challenges including insufficient depth of pathway coverage and stress-related artifacts. The liquid diet method reported here is an effective and noninvasive solution to these long-standing problems. The method can be readily extended to animal model studies using other labeled fuel sources. Our data also revealed complex metabolic interactions between tumors and the host, that is challenging to resolve in the present study. This will require the development of a multicompartment dynamic flux model using positional isotopomer and mass isotopologue data collected at multiple time points (cf. Fig. 2) to better and quantitatively account for the metabolic complexity of in vivo systems[33, 34].

## Methods

**Animals.** The NOD/SCID gamma (NSG) mouse colony was maintained in the Division of Laboratory Animal Resources of the University of Kentucky. Initial breeding pairs were purchased from The Jackson Laboratory (Bar Harbor, ME). Mice were housed in a climate-controlled environment with a 1410 hours light/dark cycle (lights on at 0600 hours) with water and solid diet (except during tracer administration via liquid diet, see below) provided ad libitum throughout the study. The institutional animal care and use committee at University of Kentucky has approved all of the animal procedures carried out in this study under PHS Assurance #A3336-01.

**$^{13}$C-glucose labeling of NSG mice.** A liquid diet base containing casein, L-cystine, soy oil, cellulose, mineral mix (AIN-93G-MX), calcium phosphate, vitamin mix (AIN-93-VX), Choline Bitartrate, tert-butylhydroquinone (TBHQ), and Xanthan gum was purchased from Harlan Laboratories (Madison, WI). S&P grade $^{13}$C$_6$-glucose was obtained from Cambridge Isotope Laboratories (Tewkebury, MA).

For the tracer study, unlabeled glucose and water were added to the diet base two days prior to the tracer study to give a final diet of 0.167 g glucose/g diet and a net protein content of 53 mg/g diet to provide sufficient carbon and nitrogen according to the vendor. Twenty g mice were fed 13.6 g liquid diet (at 680 g diet/kg mouse). This pre-feeding of unlabeled liquid diet served to accustom the mice to the liquid diet feeding. On the third day, $^{13}$C$_6$-glucose at 0.173 g/g diet replaced the unlabeled glucose in the diet for each mouse and the mice were allowed to consume the diet ad libitum for 18 h. At the end of the feeding period, mice were sacrificed by spinal dislocation and organs were harvested and snap frozen in liquid nitrogen.

The frozen tissues were pulverized into 10 µm particles in liq. N$_2$ using a Spex freezer mill (Spex). 300 mg of each pulverized tissue were extracted in acetonitrile/water/chloroform (V/V 2:1.5:1) and separated into polar, lipid, and protein fractions[35].

**Cell culture and $^{13}$C-glucose labeling in vitro.** An arsenite-transformed BEAS-2B (BAST) lung epithelial cell line[36] was maintained in DMEM supplemented with 10 mM glucose, 2 mM Gln, 10% heat-inactivated FBS and 1% penicillin–streptomycin

in 10 cm dishes. For the tracer experiment, BAST cells were allowed to reach ~50% confluency, followed by the addition of DMEM base media supplemented with 10 mM $^{13}$C$_6$-glucose, 2 mM Gln, 10% dialyzed fetal bovine serum, 50 U/mL penicillin, and 50 µg/ml streptomycin for 24 h in a CO$_2$ incubator maintained at 37 °C. The enrichment in $^{13}$C$_6$-glucose was > 99%. At the end of incubation, cells were washed with cold PBS three times followed by extraction with acetonitrile/water/chloroform (V/V 2:1.5:1) and separated into polar, lipid, and protein fractions[35].

**Human lung cancer slices and $^{13}$C-glucose labeling ex vivo.** Fresh non-small cell lung cancer specimens were obtained at surgery from a diagnosed, early-stage patient (UK025) in March 2015. Written informed consent was obtained from the patient and the study was approved by the University of Kentucky Internal Review Board under 14-0288-F6A.

Fresh cancerous (CA) and surrounding non-cancerous (NC, distal and proximal to CA) lung tissues were collected at the operating room within 5 min of surgical resection, tissues were thinly sliced by the surgeon at ~0.7–1 mm thickness with net weights of ~8–30 mg tissue slices, as previously described[37, 38]. All tissue slices were incubated in glucose and Gln-free DMEM media (Sigma-Aldrich, St. Louis, MO) supplemented with 0.45% unlabeled or $^{13}$C$_6$-glucose (Cambridge Isotope Laboratories, Tewksbury, MA), 2 mM Gln, 10% dialyzed fetal bovine serum, 50 U/mL penicillin, and 50 µg/mL streptomycin (Fisher Scientific, Waltham, MA) for 24 h at 37 °C/5% CO$_2$ with gentle rocking to facilitate nutrient supply and waste product dilution[37, 38]. At the end of incubation, tissue slices were quickly rinsed in cold PBS 3 times to remove medium components, blotted dry, weighed on a 4- place balance for wet weight, and flash-frozen in liquid N$_2$. The frozen tissues were pulverized to 10 µm particles in liq. N$_2$ using a Spex freezer mill (Spex). 20 mg of each pulverized tissue were extracted in acetonitrile/water/chloroform (V/V 2:1.5:1) and separated into polar (aqueous layer), lipid (chloroform layer) and protein (interfacial layer) fractions[35].

**Mouse PDTX Models.** Surgical tumor samples were cut into ca. 2 mm pieces and transplanted within 30 min s.c. into two immunodeficient NSG mice at both left and right flanks. The gender of the mice was chosen to match that of the donor patient. Mice were observed daily for tumor growth. When the tumor reached about 1 cm$^3$ in size, both mice were fed $^{13}$C$_6$-glucose enriched liquid diet for 18 h before necropsy. A portion of the tumors were removed immediately and snap frozen in liquid N$_2$, while the rest of the tumors were cut into 2 mm pieces and re-implanted into NSG mice for propagation. The frozen tumors were pulverized and extracted as described above.

**Cell line xenograft model.** Arsenite-transformed BEAS-2B (from ATCC, certified and mycoplasma free) (BAsT) NSCLC cells[36] were suspended in PBS (5 × 10$^6$ in 100 µl) and injected subcutaneously at both left and right flanks of NSG mice. Tumors were allowed to develop to 1 cm$^3$ in size. Mice were then fed the $^{13}$C$_6$-glucose enriched liquid diet for 18 h before necropsy. A blood sample was taken from the submandibular vein into a K$_3$-EDTA blood tube for plasma isolation before the mice were killed by cervical dislocation. Tumors and other tissues were removed immediately after euthanizing and snap frozen in liquid N$_2$. The frozen tumors were pulverized and extracted as described above.

**Polar metabolite analysis by NMR spectroscopy.** NMR spectra were recorded at 14.1 T and 15 °C on a Varian DD2 spectrometer equipped with a 3 mm inverse triple resonance cold probe. 1D $^1$H NMR spectra were recorded with an acquisition time of 2 s, a recycle time of 6 s to minimize peak saturation, and continuous rf presaturation of the residual water resonance at 4.88 ppm. 1D $^1$H-{$^{13}$C} HSQC spectra were recorded with an acquisition time of 0.25 s, a recycle time of 2 s, and adiabatic decoupling of the $^{13}$C. Proton spectra were typically processed with zero filling to 131k points, and apodized with an unshifted Gaussian and a 0.5 Hz line broadening exponential. HSQC spectra were processed with zero-filling to 16 k points and apodized using an unshifted Gaussian function and 4 Hz line broadening. Concentrations of metabolites and $^{13}$C abundance were determined by peak integration of the $^1$H or HSQC NMR spectra referenced to the intensity of DSS-d$_6$ methyl protons, with correction for differential relaxation, as previously described[39].

**Polar metabolite analysis by IC–MS.** Polar fractions were reconstituted in 30 µl nanopure water, and analyzed on a Dionex ICS-5000+ion chromatograph interfaced to an Orbitrap Fusion Tribrid mass spectrometer (Thermo Fisher Scientific, San Jose, CA, USA) operating at a resolution setting of 500,000 (FWHM at m/z 200) on MS1 acquisition to capture any and all $^{13}$C isotopologues[40]. The Orbitrap Fusion was tuned and calibrated according to the manufacturer's default standard recommendations, to routinely achieve a mass accuracy of 0.5 ppm or better. The chromatography was outfitted with a Dionex IonPac AG11-HC-4 µm RFIC&HPIC guard (2 × 50 mm) guard column upstream of a Dionex IonPac AS11-HC-4 µm RFIC&HPIC (2 × 250 mm) column. An m/z range of 80–700 except for the organ extracts where the m/z range was 50-750. Peak areas were integrated and exported to Excel via the TraceFinder 3.3 (Themo) software package. Peak areas were corrected for natural abundance distribution of each of the isotopologues[41], after which fractional enrichment and µmoles metabolites/g protein were calculated to quantify $^{13}$C incorporation into various pathways.

**Lipid analysis by mass spectrometry**. Lipid extracts were dissolved in chloroform/methanol (2:1, v/v) containing 1 mM butylated hydroxytoluene (BHT) as antioxidant and subjected to MS analysis using an Orbitrap Fusion interfaced to a TriVersa NanoMate (Advion Biosciences, Ithaca, NY, USA) nanoelectrospray outfitted with a "D" chip (nozzle inner diameter 4.1 μm). Aliquots of lipid extracts were diluted 10-fold in solvent A (2-propanol/methanol/chloroform, 4:2:1, v/v/v, containing 20 mM ammonium formate) and B (methanol) for positive and negative ion mode analysis, respectively. Samples were placed into cleaned 96-well plates, sealed with PTFE-lined seals (type BST-9790, Laboratory Supply Distributors Corp, Vineland, NJ, USA), and the plate kept at 4 °C during automated runs on the Nanomate to reduce sample evaporation/condensation in the sealed wells, as well as to maintain analyte stability. Nanoelectrospray ionization was initiated from the nozzle by applying 1.6 kV with a 0.6 psi head pressure in positive ion mode, and 1.5 kV and a 0.5 psi head pressure in the negative ion mode. All UHR-FTMS data were recorded in profile mode using a maximal injection time of 100 ms, automated gain control at $2.5 \times 10^5$, five microscans, and a target resolution of 500,000 (FWHM at m/z 200). The Orbitrap Fusion was tuned and calibrated according to the manufacturer's default standard recommendations, to routinely achieve a mass accuracy of 0.5 ppm or better. Assignments for various lipid classes and their isotopologue distributions were determined using our in-house m/z-alignment & matching software Precalculated Exact Mass Isotopologue Search Engine (PREMISE)[42]. The fractional $^{13}$C enrichment in lipid species was calculated after natural abundance correction[41, 42]. Isotopologue distributions were shown for different lipid classes as the unlabeled fraction (m0), the fraction in which only the glycerol subunit was enriched at all three positions (m3), the fraction in which only the acyl chains were enriched (summed acetyl units, "even") and the fraction in which both the glycerol and the acyl chains were $^{13}$C-enriched (odd > 3)[2, 42].

**Protein hydrolysis and assignment by mass spectrometry**. The extracted protein was precipitated with 10% trichloroacetic acid (TCA), then hydrolyzed into free amino acids in 6N HCl using a focused beam microwave (CEM Discover) at 160 °C for 10 min. Subsequently, samples were freeze-dried to remove HCl, followed by ethyl chloroformate (ECF) derivatization, as previously described[43]. Following derivatization, ECF derivatives were introduced into the Orbitrap Fusion by direct nanoelectrospray using the Nanomate (operated at 1.5 kV and 0.5 psi head pressure for positive mode). The maximum ion time for the automatic gain control (AGC) on the Orbitrap was set to 100 ms. Five transients were added to produce each stored spectrum. Spectra were acquired for 15 min.

**Statistical analyses**. Metabolite quantifications are given as mean ± S.E.M. For isotope enrichments (i.e., above natural abundance), significance was tested with a one tailed $t$-test with the null hypothesis of no enrichment. With the measured SEM and $n = 3$, we determined that the minimum enrichment that reaches statistical significance was 0.02–0.05. For comparing means a two-tailed unpaired $t$-test was used. For comparing enrichments in organs a paired $t$-test was used, corrected for false discovery, as described in refs. [44, 45]. Variances were close enough to justify the $t$-test assuming equal variance. A $P$ value of < 0.05 was considered significant. *$0.01 < P < 0.05$; **$0.001 < P < 0.01$; ***$P < 0.001$. Statistics were calculated using either GraphPad or Kaleidagraph (Synergy software).

**Data availability**. Metabolic data are available from the authors upon request and will be made available to the community via the Metabolomics Workbench (www.metabolomicsworkbench.org). The authors declare that all the other data supporting the findings of this study are available within the article and its Supplementary Information files and from the corresponding author upon reasonable request.

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

## Acknowledgements

This work was supported in part by grants: 1R01ES022191-01 (to T.W.-M.F. and R.M. H.), 1P01CA163223-01A1 (to A.N.L. and T.W.-M.F.), 1U24DK097215-01A1 (to R.M.H., T.W.-M.F., and A.-N.L.), 5R21ES025669-02 (to T.W.M.F.) and the Redox Metabolism Shared Resource(s) of the University of Kentucky Markey Cancer Center (P30CA177558). A.T.L. was supported by T32 5T32CA160003-05 (B.M.E.). R.C.S. was supported by a T32 training grant 5T32ES007266-25 (M.V.). ¹³C-enriched standards were obtained from NIH Common Fund Metabolite Standards Synthesis Core (http:// www.metabolomicsworkbench.org/standards/index.php). We would also like to thank Ms. Yan Zhang and Teresa Cassel for helping with the mouse necropsy and the BAsT xenograft experiments, Dr Christine Brainson for consulting on mouse pathology and histology, and Dr Piotr Dobrowolski for acquiring the NMR data.

## Author contributions

R.C.S. designed and performed the experiments, analyzed NMR and MS results, and co-wrote the manuscript. Q.S., M.O.W., R.M.H., contributed to data acquisition, performed part of the IC-UHR-FTMS data analysis, and wrote the method section on IC-FTMS. M.O.W. established the IC-FTMS data analysis procedure and helped IC-FTMS data curation, Y.Y. performed the protein hydrolysis and UHR-FTMS analysis, and wrote the protein hydrolysis method. P.D. performed lipids analysis by UHR-FTMS. A.-T.L. helped perform mouse necropsies and extractions. T.L.S. performed UHR-FTMS analysis on plasma glucose. A.N.L. helped perform the mouse necropsies, statistical analysis and interpret data. T.W.M.F. conceptualized the liquid diet method, contributed to experimental design, SIRM data and statistical analysis, and bulk of the metabolic interpretation as well as co-wrote the manuscript.

## Additional information

**Competing interests:** The authors declare no competing financial interests.

**Reprints and permission** information is available online at http://npg.nature.com/ reprintsandpermissions/

