## [Peer Review File · Nature Communications]

Reviewer #1 (Remarks to the Author):

In this study, Sun et al. demonstrate that administration of ^{13}C glucose via a liquid diet for 18 hours leads to significant enrichment in a broad range of metabolite pools as well as protein. Isotope tracer administration for probing flux in vivo is of immense interest to the field, as the technical challenges and physical stress associated with cannulation and infusion (the gold standard approach) are problematic for investigators. Validation of alternative approaches such as this is of importance for future in vivo metabolic studies using stable isotopes. Key strengths of this study are the deep coverage of metabolic pathways achieved via utilization of multiple analytical platforms as well as the feeding pattern assessment of the mice. However the interest and impact would increase if kinetic information were provided for glucose labeling. Alternatively, direct comparison of this approach with other means of ^{13}C glucose administration would strengthen this study.

Major comments:

The authors state that a benefit of this approach is the high reproducibility however 3 or less mice in one biological context/grouping were used for the studies presented. This makes it hard to assess/make claims about the reproducibility of the approach or whether it is sensitive enough to detect differences between groups of mice.

As noted above, some kinetic measurements to establish whether steady state (or pseudo-steady state) glucose labeling is established (or not, which may be fine).

Minor comments:

Is calorie consumption per day altered when you switch from normal chow to liquid diet?

Figure 1 B and C are switched relative to what's described in the text. 4PM is included twice in a 24 hour timeline rather than 4 AM.

Please label the axes of graphs presented in the supplementary information.

Table S1. Present macronutrient composition as % kcal as well as g/kg.

In figure S9 C, please reduce the number of digits following the decimal place.

Reviewer #2 (Remarks to the Author):

In this manuscript, Sun et al. traced the fate of ^{13}C , which is introduced by [U- $^{13}\text{C}_6$] glucose enriched liquid diet, in several mice tissues and xenografts. As the title suggests, an indisputable benefit of ^{13}C liquid diet is noninvasive delivery of tracers; however, there are disadvantages to this isotope delivery method. The disadvantages include: 1) inability to study the fasting state; 2) long time to substantial incorporation of ^{13}C into cellular metabolites; 3) as a result of long labeling, metabolite/macromolecule labeling data represent a compound effect of disparate physiology; 4) the question of isotopic pseudo-steady state, which facilitates well-established metabolic flux analysis; and 5) if not at pseudo-steady state, complications in quantitative analysis of isotope labeling data. If the authors had addressed the challenges associated with these disadvantages (especially, 3), 4) and 5)), their work could have contributed substantially to understanding the whole body metabolism and in vivo cancer metabolism.

The present work employed suitable analytical tools (i.e., high resolution mass spectrometry and HSQC NMR) for measuring soluble metabolites, proteinogenic amino acids, glycogen and lipids. Understanding their results and assessing the validity of the conclusions were encumbered by poor presentation of the data. There was not a substantial result or finding, and the main text and figures failed to convey the importance of the present work.

Minor points:

Figure 1: The panels B and C need to be switched as they don't match figure legends. The last sentence of Fig. 1 legend is confusing and irrelevant to the figure.

Figure 2 seems to be a fair starting point for quantitative analysis of isotope labeling data obtained in this study. Assessing how well the labeling data can be explained by the set of reactions in this figure may add value to the study.

We would like to thank the reviewers for their thoughtful and generally favorable comments on the manuscript, in particular about the critiques on reaching isotopic steady state and reproducibility of liquid diet to enhance the scientific merit of this manuscript. We have performed three additional experiments to address all of the comments and concerns raised by both reviewers, including a time course tracer measurement and one each biological comparison of a 2D human transformed cell line or *ex vivo* human tissue culture versus the corresponding mouse xenografts, as well as made corrections to the manuscript to improve readability.

Major revisions to the manuscript are outlined as follows.

(1) We have added a time course tracer experiment in the mice to determine the approach to isotopic steady state as shown in the new **Figure 2.**

(2) We have added **Figure S1** to demonstrate reproducibility by (1) overlaying NMR spectra of extracts of several organs from different mice, and (2) a dot plot of total amount of ^{13}C -enriched malate, succinate, citrate, fructose-6-phosphate, glucose-6-phosphate, and phosphoenolpyruvate from 10 mouse livers with the corresponding coefficient of variation (CoV).

(3) We have compared the metabolism of patient-derived xenografts (PDX) *in vivo* with fresh *ex vivo* lung tissue slice cultures from the same patient to map the influence of tumor microenvironment on metabolic networks by comparing the two tumor models. These are presented as **Figs. 3, S7-S9.**

(4) We have compared the metabolism of an arsenite-transformed human bronchial epithelial (BA5T) cell line in mouse xenografts (*in vivo*) versus the 2D cell culture (*in vitro*) to illustrate again the influence of tumor microenvironment on metabolic networks. This is now presented as **Figure 4.**

We would also like to emphasize that we developed this liquid diet method to circumvent the limitations of current *in vivo* labeling methods (bolus injection and cannulation) to achieve much deeper metabolic coverage while avoiding overt stress on the animals. We hope that the revised manuscript is now acceptable for publication in Nature Communications.

Please find specific responses to the reviewers' comments below.

Reviewer #1 (Remarks to the Author):

*"In this study, Sun et al. demonstrate that administration of ^{13}C glucose via a liquid diet for 18 hours leads to significant enrichment in a broad range of metabolite pools as well as protein. Isotope tracer administration for probing flux *in vivo* is of immense interest to the field, as the technical challenges and physical stress associated with cannulation and infusion (the gold standard approach) are problematic for investigators. Validation of alternative approaches such as this is of importance for future *in vivo* metabolic studies using stable isotopes. Key strengths of this study are the deep coverage of*

metabolic pathways achieved via utilization of multiple analytical platforms as well as the feeding pattern assessment of the mice. However the interest and impact would increase if kinetic information were provided for glucose labeling. Alternatively, direct comparison of this approach with other means of ^{13}C glucose administration would strengthen this study. “

We thank the reviewer for the favorable comments.

Major points:

1. *“The authors state that a benefit of this approach is the high reproducibility however 3 or less mice in one biological context/grouping were used for the studies presented. This makes it hard to assess/make claims about the reproducibility of the approach or whether it is sensitive enough to detect differences between groups of mice.”*

We agree that three measurements did not adequately sample the variability. We have therefore acquired and presented the mean and covariance of fractional enrichments in six metabolites extracted from the livers of 10 mice from 3 separate liquid diet experiments, as shown in Figure S1 of the revised manuscript. The high reproducibility of this approach is also illustrated in the new Figure 2, which showed the time course analysis of tracer administration via liquid diets. Furthermore, we demonstrated the reproducibility by comparing metabolite labeling between PDX vs. *ex vivo* tumor slices of the same patient (Figure 3) and BASt 2D cell line versus xenografts (Figure 4) to show that this approach is sufficiently robust to detect differences between each set of tumor models.

2. *“As noted above, some kinetic measurements to establish whether steady state (or pseudo-steady state) glucose labeling is established (or not, which may be fine).“*

We performed a time course experiment and determined ^{13}C enrichment in metabolites of plasma and livers in mice fed with $^{13}\text{C}_6$ -glucose based liquid diet at t=0, 12, 24, 36 and 48 hours. We found that the majority of metabolites reached isotopic steady states by 12 hours of *ad libitum* feeding, whereas a few took longer to approach isotopic steady state, as described in the text (pp. 3-4) and the new Figure 2.

Minor points:

1. *“Is calorie consumption per day altered when you switch from normal chow to liquid diet?”*

We have examined this point, and found there was no measurable difference in calorie consumption. The liquid diet was formulated to provide the same numbers of calories from carbohydrates, proteins and fats as the solid chow.

2. *“Figure 1 B and C are switched relative to what’s described in the text. 4PM is included twice in a 24 hour timeline rather than 4 AM.”*

We have done the following. Experimental design has been added to Fig. 1B and mouse feeding pattern is now Fig. 1C. Liver NMR spectra have been moved to Fig. S1B 4 pm is corrected to 4 am on the mouse feeding pattern.

3. *“Please label the axes of graphs presented in the supplementary information.”*

We have added “fraction” to the Y-axis and stated that X-axes are “mass isotopologues” in the corresponding figure legends. We have done this for both main and supplemental figures.

4. *“Table S1. Present macronutrient composition as % kcal as well as g/kg.”*

We added %kcal as an extra column to the table and moved the table to the main text as Figure 1A.

5. *"In figure S9 C, please reduce the number of digits following the decimal place.*

Fig. S9 is now Fig. S10 and Fig. S9C has been removed as it did not contribute much to the argument. Instead, we added Fig. S10C to show the scrambled patterns of plasma ^{13}C -glucose for comparison with those of tumor ^{13}C -F1,6P (Fig. 4A) for BASt cell xenograft mice.

Reviewer #2 (Remarks to the Author):

"In this manuscript, Sun et al. traced the fate of ^{13}C , which is introduced by [U- $^{13}\text{C}_6$] glucose enriched liquid diet, in several mice tissues and xenografts. As the title suggests, an indisputable benefit of ^{13}C liquid diet is noninvasive delivery of tracers; however, there are disadvantages to this isotope delivery method. The disadvantages include: 1) inability to study the fasting state; 2) long time to substantial incorporation of ^{13}C into cellular metabolites; 3) as a result of long labeling, metabolite/macromolecule labeling data represent a compound effect of disparate physiology; 4) the question of isotopic pseudo-steady state, which facilitates well-established metabolic flux analysis; and 5) if not at pseudo-steady state, complications in quantitative analysis of isotope labeling data. If the authors had addressed the challenges associated with these disadvantages (especially, 3), 4) and 5)), their work could have contributed substantially to understanding the whole body metabolism and in vivo cancer metabolism."

We thank the reviewer for raising some important issues to clarify. We consider that the dietary approach has valuable advantages over other means of tracer administration, but it does not replace other methods. The method of choice depends on the questions being posed (p. 8). We have made a number of changes to the manuscript to address these questions.

1. *"inability to study the fasting state."*

Direct measurement of metabolic processes in the fasting state is indeed challenging with the liquid diet method. However, the liquid diet method can be readily employed to study metabolic activity during refeeding. We demonstrated such application in the revised manuscript by fasting mice for 6 hours (3 am-9 am) before feeding with the $^{13}\text{C}_6$ -glucose enriched liquid diets in a time course experiment for examining the approach to isotopic steady state of different metabolites (Fig. 2). The fasting period can be manipulated as desired. Other means of administering tracers (e.g. continuous infusion) also do not accurately measure metabolic activity in the fasting state, due to the influence of the infused nutrient(s).

An alternative means to measure metabolism, albeit a more costly approach, during fasting is to extensively pre-label the animals by liquid diet, followed by tracking the loss of ^{13}C during the period of fasting (a pulse-chase method).

2. *“long time to substantial incorporation of ^{13}C into cellular metabolites.”*

We agree that for very short-term tracer studies (e.g. 15 min to a few hours), the liquid diet method may not be the method of choice and was not the intended purpose. We are particularly interested in alleviating stress during prolonged tracer administration and in enabling tracking transformations of metabolites with slow turnover rates (e.g. nucleotides, lipids, and proteins), which have been difficult to capture with existing short-term (up to 3-4 h) tracer administration methods. As such, our method complements the existing methods as we can reliably track mouse metabolic activity after 12 hr of *ad libitum* feeding of labeled tracers (shorter durations may still be feasible). At the same time, it does not incur overt stress to confound metabolic activity.

3. *“as a result of long labeling, metabolite/macromolecule labeling data represent a compound effect of disparate physiology.”*

We acknowledge that this is a possibility. The fact that we obtained highly reproducible metabolite labeling patterns with multiple mice using the liquid diet method (cf. Figs. 2, S1) suggests that the resting physiological state of individual mice was highly comparable, which we presume was largely due to the stress-free conditions. Consequently, we were able to demonstrate the influence of human versus mouse TME (Fig. 3) or *in vitro* versus *in vivo* conditions (Fig. 4) on tumor metabolism.

4. *“the question of isotopic pseudo-steady state, which facilitates well-established metabolic flux analysis.”*

As stated in the response to Reviewer 1 point #2, the majority of metabolites reached isotopic steady states after 12 hours of *ad libitum* feeding (cf. Figure 2). Nevertheless some metabolites required longer periods to reach isotopic steady state, which indicates the need for long-term tracer administration for steady-state based flux modeling purposes. The liquid diet method that we introduced in this report will facilitate such research.

5. *“if not at pseudo-steady state, complications in quantitative analysis of isotope labeling data.”*

As stated above in point #4, we now demonstrated that isotopic steady-state was reached within 12 hr of tracer feeding for most metabolites, which should enable quantitative analysis through flux modeling methods that require isotopic steady state.

For nucleotides such as ATP (Fig. 2H), longer than 48 hr of tracer labeling may be required to reach isotopic steady state, which will be enabled by the liquid diet method.

6. *“The present work employed suitable analytical tools (i.e., high resolution mass spectrometry and HSQC NMR) for measuring soluble metabolites, proteinogenic amino acids, glycogen and lipids. Understanding their results and assessing the validity of the conclusions were encumbered by poor presentation of the data. There was not a substantial result or finding, and the main text and figures failed to convey the importance of the present work.”*

We apologize for the errors in presentation. We have carefully matched the figures to the text, and revised the text to better convey the importance of our work. In addition, we performed three more experiments to illustrate the potential for applications in 1) steady-state flux modeling by showing the ability to reach isotopic steady-state (Fig. 2);

2) metabolic studies of mouse models of human cancer cells including patient-derived tumor xenografts and transformed cell xenografts in comparison with other models (*ex vivo* tumor tissue or 2D cell cultures) (Figs. 3 and 4). From these comparisons, we found distinct features of human cancer cell metabolism *in vitro*, *ex vivo*, and *in vivo* as mouse xenografts that could be attributed to the influence of TME. We believe that the liquid diet method will enable a much wider range of applications to probe metabolic reprogramming in xenograft, syngeneic, and transgenic animal models of human diseases.

Minor points:

1. *“Figure 1: The panels B and C need to be switched as they don’t match figure legends. The last sentence of Fig. 1 legend is confusing and irrelevant to the figure.”*

We have done the following. Experimental design has been added to Fig. 1B and mouse feeding pattern is now Fig. 1C. Liver NMR spectra have been moved to Fig. S1B 4 pm is corrected to 4 am on the mouse feeding pattern. Fig. 1 legend has been adjust accordingly.

2. *“Figure 2 seems to be a fair starting point for quantitative analysis of isotope labeling data obtained in this study. Assessing how well the labeling data can be explained by the set of reactions in this figure may add value to the study.”*

We agree that these data would be valuable for a quantitative flux analysis, which was an incentive for developing the liquid diet method for tracer studies. However, proper flux modeling (even with the assumption of isotopic steady state) is complicated by the need to establish multicompartiment (across organs and subcellular organelles) models to assess fluxes accurately, which is a development that we are planning in the future but is beyond the scope of this paper.

Reviewer #1 (Remarks to the Author):

The added time course tracer data and xenograft comparisons provide important additional data for assessing the utility of this approach.

Minor points:

Include details of isotope tracing in ex vivo slices and cell culture in materials and methods.

In the second sentence of discussion, it is claimed that the demonstrated advantages of this approach include mitigation of metabolic interferences resulting from restraint stress etc. While this method potentially mitigates these factors, this study does not demonstrate this as no direct comparison was made with other methods. Please change wording of this claim.

Reviewer #2 (Remarks to the Author):

In this revised manuscript, Sun et al. described and applied ^{13}C -enriched liquid diet to understand mouse in vivo metabolism and metabolism in xenografts. The authors addressed many of the issues and overall improved the relevance of their work. Some major issues remain:

I mentioned in the previous review that "as a result of long labeling, metabolite/macromolecule labeling data represent a compound effect of disparate physiology." I am still not convinced that the long-term labeling associated with ^{13}C -enriched liquid diet can capture the differences in metabolism in different physiological states. To put it another way and for example, can the differences of metabolism in active and resting states be reflected in metabolite labeling from ^{13}C liquid diet?

In Fig. 2, showing total ^{13}C enrichment is not sufficient; please show mass isotopomer distributions (M+0, M+1, etc) for those metabolites. Furthermore, please explain why the error bars here seem smaller than those of the same metabolites in other figures (3, 4, and Supp. Figs.).

While it is expected that different microenvironments/culture conditions affect cellular metabolism, the authors' interpretation of the labeling data (in Figs. 3, 4, S7-S10) and conclusions are misleading. For example, with most glycolytic and PPP metabolites nearly uniformly- ^{13}C labeled in tissue slices, I do not understand how the authors can get any information about their pathway usage. In general, the observed differences in the labeling of PDX and tissue slices may be mostly explained by the differences in the glucose and other circulating metabolite labeling between mouse serum and in ex vivo culture medium. The same is true for in vivo vs. in vitro BAsT cells. The ex vivo/in vitro tissue culture conditions need to be described and the differences in the circulating "input" metabolite labeling must be accounted for in interpreting the downstream metabolite labeling.

Minor issues:

In Introduction: '2D cell lines' is confusing and should be reworded.

Reviewer #1 (Remarks to the Author):

1. *Minor points: "Include details of isotope tracing in ex vivo slices and cell culture in materials and methods."*

This has been added to the Online method section (referenced on p10).

2. *"In the second sentence of discussion, its claimed that the demonstrated advantages of this approach include mitigation of metabolic interferences resulting restraint stress etc. While this method potentially mitigates these factors, this study does not demonstrate this as no direct comparison was made with other methods. Please change wording of this claim."*

We have re-worded this statement (p. 9).

Reviewer #2 (Remarks to the Author):

1. *"I am still not convinced that the long-term labeling associated with ^{13}C -enriched liquid diet can capture the differences in metabolism in different physiological states."*

To demonstrate this, we performed an 18 h liquid diet tracer experiment on PDTX mice bearing a primary tumor on the flank and metastatic lesions in their lymph nodes. These two tumor types are of the same patient origin but differ in stages and presumably physiological states. Clear differences in metabolic activities were evident between the two tumor types (cf. new **Fig. 4**). We would also like to point out that organ-specific metabolic activities were observed by NMR (**Fig S1**) and IC-MS (**Fig S2-4**) analyses, which is consistent with known differences in organ physiology (e.g. production of neurotransmitter GAB in brain (**Fig. S1D**)).

2. *"In Fig. 2, showing total ^{13}C enrichment is not sufficient; please show mass isotopomer distributions ($M+0$, $M+1$, etc) for those metabolites. Furthermore, please explain why the error bars here seem smaller than those of the same metabolites in other figures (3, 4, and Supp. Figs.)."*

We have revised **Fig. 2** and text on pp. 3-4 (Results section) to include all significant ^{13}C isotopologues of the ten example metabolites. The differences in error bars between the original **Fig. 2** and other figures are due to the differences in the fraction of the ^{13}C species plotted, i.e. total fraction of ^{13}C labeled species in **Fig. 2** versus fractions of individual ^{13}C isotopologues in other figures. In the revised **Fig. 2** where fractions of individual ^{13}C isotopologues are shown, the errors bars are now comparable to those in other figures.

3. *While it is expected that different microenvironments/culture conditions affect cellular metabolism, the authors' interpretation of the labeling data (in Figs. 3, 4, S7-S10) and conclusions are misleading. For example, with most glycolytic and PPP metabolites nearly uniformly- ^{13}C labeled in tissue slices, I do not understand how the authors can get any information about their pathway usage. In general, the observed differences in the labeling of PDX and tissues slices may be mostly explained by the differences in the glucose and other circulating metabolite labeling between mouse serum and in ex vivo culture medium. The same is true for in vivo vs. in vitro BAsT cells. The ex vivo/in vitro tissue culture conditions need to be described and the differences in the circulating "input" metabolite labeling must be accounted for in interpreting the downstream metabolite labeling.*

We agree with the reviewer in terms of the need to consider the impact of differences in the labeled precursor inputs on the labeling patterns of metabolite products in vivo, ex vivo, and in vitro. The enrichment of input $^{13}\text{C}_6$ -glucose in ex vivo and in vitro cultures was essentially 100%

(cf. revised Online Methods) while that in vivo was less than 100% (**Fig. S9**). Although the decreased fractional enrichment and increased scrambling of the $^{13}\text{C}_6$ -glucose precursor in mouse blood plasma (**Fig. S9A**) could contribute to the lower extent of ^{13}C enrichment and higher extent of ^{13}C scrambling in metabolite products in vivo versus ex vivo (**Fig. S8**), they cannot account for the enhanced abundance of ^{13}C -lactate, -Ala, -acetate, -Glu, -succinate, -glutathiones, -glycogen, and ^{13}C -ribose-nucleotides in vivo versus ex vivo (**Fig. S7A**). Likewise, the reduced extent of ^{13}C labeling in the glucose input in vivo (**Fig. S9C**) could not explain the elevated abundance (**Fig. S7B**) and enhanced fractional enrichment (**Fig. 5**) of ^{13}C -labeled Krebs cycle metabolites as well as the increased accumulation of ^{13}C -lactate, -Ala, and -glycogen (**Fig. S7B**) in vivo versus in vitro. These unaccountable differences could be attributed to microenvironmental influences including the uptake of nutrients via media or blood. We have revised the text on pp. 6-9 accordingly.

4. *Minor issues: "In Introduction: '2D cell lines' is confusing and should be reworded."*

We have deleted 2D (p 2) and changed to monolayer cell culture (p 9).

Reviewer #2 (Remarks to the Author):

The authors have addressed all my main concerns. This paper can be published after addressing the minor issues below.

A) I did not find the NMR spectra for nucleotide bases although the main text reads: "De novo synthesis of the purine and pyrimidine rings were evident from the ^{13}C signals in the 8.2-8.6 ppm region and at 8.03 ppm respectively in the lung, liver and heart spectra."

B) Page 5: Reference to Fig. '3F' in the first few sentences should be removed.

C) I was initially very confused by the mixed use of, for example, 'm3' and '3' (in page 5 and figure captions) to denote the 'M+3' isotopologue. Please be consistent and use more standard notations.

D) In Fig 4, please also show labeling incorporation as fractions of isotopologes. The following statement is incorrect if fractional labeling is considered: "lymph node lesions showed consistently higher ^{13}C incorporation from $^{13}\text{C}_6$ -glucose into the PPP (Fig. 4I), GSH (Fig. 4H), and nucleotide products (Fig. 4J and K) than the primary tumor." The authors could have made the same conclusion simply from metabolite pool size measurement, which does not require isotope labeling. Discuss the fractional labeling differences/similarities between two tumor types.

E) Please discuss the need for quantitative metabolic flux analysis by isotopomer mass balance for future studies to better interpret such complex labeling data.

Specific responses.

A) *"I did not find the NMR spectra for nucleotide bases although the main text reads: "De novo synthesis of the purine and pyrimidine rings were evident from the ^{13}C signals in the 8.2-8.6 ppm region and at 8.03 ppm respectively in the lung, liver and heart spectra."*

The 8.03-8.6 ppm region for the three organs has been included in Fig. S1 and the text (p. 4) has been revised accordingly.

B) *"Page 5: Reference to Fig. '3F' in the first few sentences should be removed."*

Reference to Fig. 3F has been removed.

C) *"I was initially very confused by the mixed use of, for example, 'm3' and '3' (in page 5 and figure captions) to denote the 'M+3' isotopologue. Please be consistent and use more standard notations."*

We have unified all notations for mass isotopologues by changing the X-axis label in Fig. S5. We use $^{13}\text{C}_i$ notation in the text for denoting isotopologues with i number of ^{13}C . Due to space issue, the X-axes were labeled by the i number instead of M+i.. We have defined this in the text and figure legends.

D) *"In Fig 4, please also show labeling incorporation as fractions of isotopologues. The following statement is incorrect if fractional labeling is considered: "lymph node lesions showed consistently higher ^{13}C incorporation from $^{13}\text{C}_6$ -glucose into the PPP (Fig. 4I), GSH (Fig. 4H), and nucleotide products (Fig. 4J and K) than the primary tumor." The authors could have made the same conclusion simply from metabolite pool size measurement, which does not require isotope labeling. Discuss the fractional labeling differences/similarities between two tumor types."*

As reviewer 2 suggested, we have added glucose- ^{13}C label incorporation as fractions in Fig. S11. We also discussed the differences/similarities between the two tumor types based on the ^{13}C pool size (in $\mu\text{mole/g}$ protein) and fractional enrichment of representative pathway products (pp. 7-8). Both parameters are needed to inform the capacity of glucose-fueled pathways. Steady-state unlabeled metabolite pool sizes cannot unequivocally provide this information since given metabolites can be derived from multiple fuel sources via different pathways. For example, glutamate can come from glutamine oxidation via glutaminolysis, instead of originating from glucose via glycolysis + the Krebs cycle. In addition, unlabeled metabolite pool sizes represent the sum of synthesis and degradation, and increase in the pool size of unlabeled ATP, for example, does not necessarily mean that its de novo synthesis is enhanced; it could reflect decreased degradation.

E) *"Please discuss the need for quantitative metabolic flux analysis by isotopomer mass balance for future studies to better interpret such complex labeling data."*

We have added this in the discussion section (p. 10).

Reviewer #2 (Remarks to the Author):

The authors have addressed all my concerns.

Responses to Referee Comments.

REVIEWERS' COMMENTS:

Reviewer #2 (Remarks to the Author):

The authors have addressed all my concerns.

We are pleased that we have now satisfied the reviewers' comments.